# SegResMamba: An Efficient Architecture for 3D Medical Image Segmentation

**Badhan Kumar Das**[1,2]                                    BADHAN.DAS@FAU.DE
**Ajay Singh**[2]                                            AJAY.SINGH@FAU.DE
**Saahil Islam**[1,2]                                        SAAHIL.ISLAM@FAU.DE
**Gengyan Zhao**[3]                         GENGYAN.ZHAO@SIEMENS-HEALTHINEERS.COM
**Andreas Maier**[2]                                       ANDREAS.MAIER@FAU.DE

[1] *Siemens Healthineers AG*
[2] *FAU Erlangen Nuremberg*
[3] *Siemens Medical Solutions USA, Inc.*

**Editors:** Accepted for publication at MIDL 2025

## Abstract

The Transformer architecture has opened a new paradigm in the domain of deep learning with its ability to model long-range dependencies and capture global context and has outpaced the traditional Convolution Neural Networks (CNNs) in many aspects. However, applying Transformer models to 3D medical image datasets presents significant challenges due to their high training time, and memory requirements, which not only hinder scalability but also contribute to elevated $CO_2$ footprint. This has led to an exploration of alternative models that can maintain or even improve performance while being more efficient and environmentally sustainable. Recent advancements in Structured State Space Models (SSMs) effectively address some of the inherent limitations of Transformers, particularly their high memory and computational demands. Inspired by these advancements, we propose an efficient 3D segmentation model for medical imaging called SegResMamba, designed to reduce computation complexity, memory usage, training time, and environmental impact while maintaining high performance. Our model uses less than half the memory during training compared to other state-of-the-art (SOTA) architectures, achieving comparable performance with significantly reduced resource demands.

**Keywords:** Mamba, State Space Models, Vision Transformer, Medical Image Segmentation

## 1. Introduction

The Transformer architecture has revolutionized deep learning by effectively modeling long-range dependencies and capturing global context. However, its application to 3D medical imaging datasets presents significant challenges, including high memory requirements, computational complexity, and prolonged training times. These challenges are particularly pronounced in tasks involving large datasets like BraTS (Baid et al., 2021) and BTCV Segmentation (Landman et al., 2015), where training Transformer-based models such as UNETR (Hatamizadeh et al., 2021b) and SwinUnetr (Hatamizadeh et al., 2021a) demands substantial resources. Furthermore, transformer models often struggle with smaller datasets, such as Spleen Segmentation (Antonelli et al., 2022), where their performance is suboptimal. The environmental impact of Transformers, driven by their elevated training times, has

raised concerns. This has led to a growing interest in alternative architectures such as structured state space models (SSMs), which reduce computational demands and training time, offering a more efficient solution for medical image analysis.

State-space architectures like Mamba (Gu and Dao, 2024), S4 (Gu et al., 2022), and S4nd (Nguyen et al., 2022) have gained popularity due to their solid foundation in Kalman Filters (Kalman, 1960). In contrast, CNN-based models like U-Net (Ronneberger et al., 2015) and SegResNet (Myronenko, 2018) are effective but have a limited receptive field. Hybrid models like UNETR and SwinUnetr(Hatamizadeh et al., 2021a) combine CNNs and Transformers(Vaswani et al., 2023) to enhance performance, though Transformers remain computationally demanding, limiting their practicality in resource-constrained clinical settings. Numerous studies have adapted Mamba to address this issue by modeling long-range dependencies with innovative selection mechanisms (Zhu et al., 2024; Wang et al., 2024; Liu et al., 2024; Liao et al., 2024; Wang and Ma, 2024).

3D image segmentation methods, such as U-Mamba(Ma et al., 2024) and SegMamba(Xing et al., 2024), leverage hybrid CNN-SSM blocks to combine the local feature extraction capabilities of convolutions with the ability of SSMs to capture long-range dependencies. Inspired by these models, we propose SegResMamba, which uses the benefits of Mamba while further reducing memory consumption and computational requirements, thereby enhancing training efficiency. SegResMamba is a lightweight Mamba-based 3D image segmentation model that offers comparable performance to other state-of-the-art (SOTA) models while significantly increasing overall efficiency. Our approach employs Tri-orientated Mamba (ToM) to enhance long-range contextual understanding, combined with CNNs for effective local feature extraction. A convolution mamba mixed block (CMMB) efficiently captures both local and global features, starting with a convolutional bottleneck and leveraging Mamba's global modeling capabilities.

## 2. Methodology

Our model is designed to be computationally efficient while maintaining competitive performance in medical image segmentation. To achieve this, we developed a powerful encoder for efficient feature extraction, followed by a lightweight decoder to reconstruct the segmentation mask. The architecture consists of an encoder, a decoder, and skip connections between them (Ronneberger et al., 2015), as illustrated in Figure 1.

### 2.1. Encoder

To minimize the overall computational cost, particularly for the downstream decoder, we designed a powerful yet efficient encoder that extracts high-quality features. Our method integrates CNNs and Mamba blocks to effectively capture both local and global feature representations while maintaining computational efficiency. The encoder is composed of four cascaded blocks, each designed to progressively downsample spatial dimensions while extracting multi-scale hierarchical features. Each block includes Downsampling Layers, convolution and Convolution Mamba Mixed Blocksas shown in Figure 1a.

The encoder begins with an initial downsampling layer, which applies a Conv3D operation with a $7 \times 7 \times 7$ kernel, a stride of $2 \times 2 \times 2$, and padding of $3 \times 3 \times 3$. This larger receptive field allows for a comprehensive feature abstraction at the early stage. Next, a

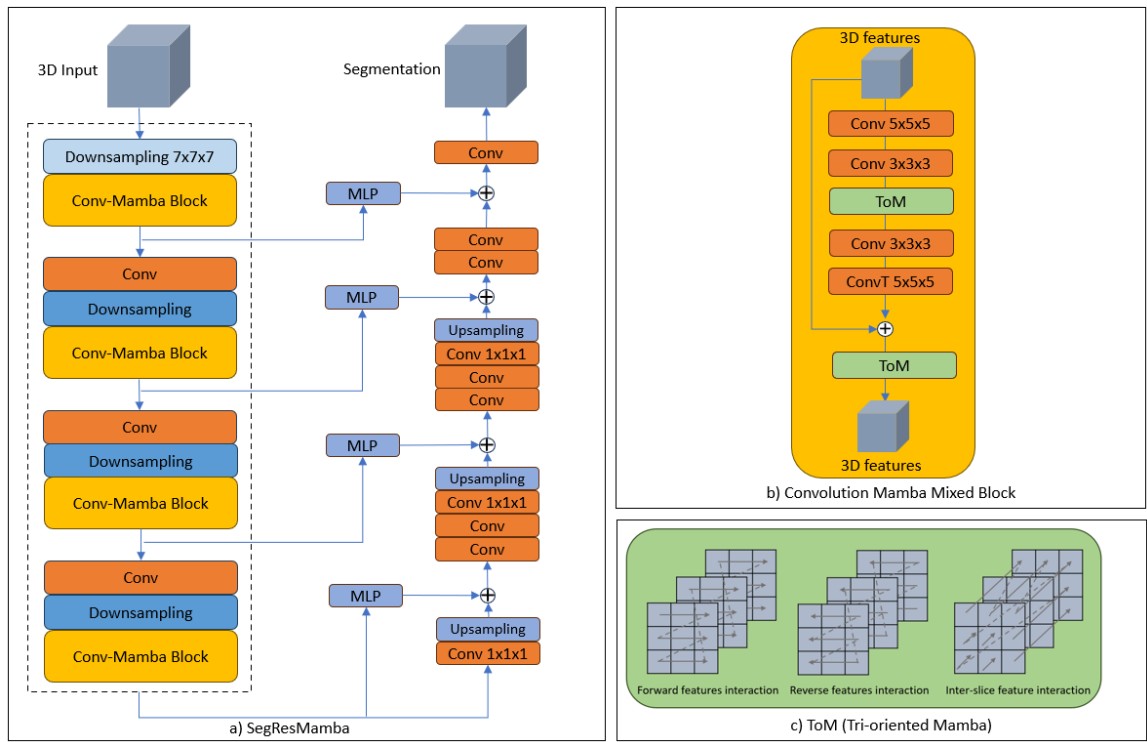

Figure 1: a) Overview of SegResMamba architecture, b) Convolution Mamba mixed block, and c) Tri-oriented Mamba

Convolution Mamba Mixed Block is applied to refine these features before further downsampling. The remaining three downsampling layers use $2 \times 2 \times 2$ Conv3D kernels, maintaining a balance between feature granularity and computational efficiency. Additionally, an extra convolutional layer is introduced before these downsampling operations to preserve essential features.

**Convolution Mamba Mixed Block:** We introduce the convolution mamba mixed block, which integrates convolutional operations and Tri-oriented Mamba (ToM) layers to achieve hierarchical feature extraction across multiple receptive fields as shown in Figure 1b.

It begins with a larger $5 \times 5 \times 5$ convolution kernel that effectively reduces the spatial dimensions while extracting coarse-grained features. These features are further refined through a $3 \times 3 \times 3$ convolution, which captures local contextual relationships.

A ToM Layer is then applied to this refined representation, enabling the abstraction of long-range dependencies and creating a more comprehensive understanding of the local context learned by the convolution filters. As shown in Figure 1c, the ToM module computes feature dependencies in three distinct directions: forward ($z_f$), reverse ($z_r$), and inter-slice ($z_s$) by flattening the 3D input features $F_2$.

$$\text{ToM}(z) = \text{Mamba}(z_f) + \text{Mamba}(z_r) + \text{Mamba}(z_s),$$

To recover the original spatial resolution, we employ a $3 \times 3 \times 3$ convolution and a $5 \times 5 \times 5$ transposed convolution. Then we have a residual connection to generate the enhanced feature representation by calculating the sum of the extracted feature map and the original feature map, which can preserve the gradient flow and retain the original feature information. Finally, a second ToM layer further captures long-range dependencies across the enhanced feature representation. The whole flow of Convolution Mamba Mixed Block is shown in Algorithm 1.

---

**Algorithm 1 Convolution Mamba Mixed Block**

---

1: **Input:** Tensor $X \in \mathbb{R}^{C \times D \times H \times W}$
2: **Output:** Feature representation $F_{\text{out}}$
3: $F_1 \leftarrow \text{Conv}_{5 \times 5 \times 5}(X)$
4: $F_2 \leftarrow \text{Conv}_{3 \times 3 \times 3}(F_1)$
5: $F_3 \leftarrow \text{ToM}(F_2)$
6: $F_4 \leftarrow \text{Conv}_{3 \times 3 \times 3}(F_3)$
7: $F_5 \leftarrow \text{ConvT}_{5 \times 5 \times 5}(F_4)$
8: $F_6 \leftarrow F_5 + X$
9: $F_{\text{out}} \leftarrow \text{ToM}(F_6)$
10: **Return** $F_{\text{out}}$

---

## 2.2. Decoder

Our decoder is intentionally designed to be lightweight compared to state-of-the-art models like UNETR, Swin-UNETR, and SegMamba, to reduce the computational cost of the model. The responsibility of maintaining the model's performance is on the uniquely designed, powerful yet efficient encoder. This efficient design reduces both computational complexity and memory usage while maintaining strong segmentation performance.

The decoder leverages both the encoded features from the encoder and the intermediate results from the encoding process. At each level, the decoder connects to the corresponding encoder layer through an MLP (Haykin, 1994) with Instance Normalization (Ulyanov et al., 2017), which normalizes activations and stabilizes training. To ensure high-resolution feature retention for precise segmentation, intermediate outputs after the MLP are directly passed to the decoder.

The decoder is structured with three distinct upsampling stages, designed to progressively refine and expand the spatial resolution of the features. The main input to the decoder has a shape of 768 in the channel dimension. At each stage, the feature map is upsampled and its channel count is halved. This process uses a $1 \times 1 \times 1$ convolution operation followed by an upsampling layer. Inside the upsampling layer, we use non-trainable linear interpolation from Monai(Cardoso et al., 2022).

Upsampled features are combined with the corresponding intermediate features received during the encoding process. Instead of concatenation, these intermediate features are

summed with the corresponding upsampled features at each level. In this way, the computational complexity of the decoder is further reduced. The combined features are processed through a sequence of residual blocks. The residual block consists of the ReLU activation function, Group Norm, and convolution kernel of $3 \times 3 \times 3$. We use two of these blocks and a skip connection from the input of these residual blocks to get an output. This architecture combines efficient upsampling with skip connections and residual learning, allowing it to reconstruct detailed spatial information while maintaining the ability to learn complex features at multiple scales. After getting the output from three decoder blocks we use a transposed convolution layer to get the final segmented output. This design is lightweight, being both memory and computation-efficient.

## 3. Experiments & Results

### 3.1. Dataset and Implementation Details

**3D Multi-organ Segmentation (BTCV Challenge):** The 3D Multi-organ Segmentation dataset from the BTCV Challenge (Landman et al., 2015) focuses on the segmentation of 13 abdominal organs. The dataset comprises 30 volumetric images, with 24 volumes allocated for training and the remaining 6 reserved for testing and evaluation. Each volumetric image provides detailed 3D representations of abdominal structures, essential for medical imaging and diagnosis. The task involves accurately delineating each of the 13 specified organs within these scans.

**BraTS 2021:** The BraTS 2021 dataset(Baid et al., 2021) comprises 1,251 multiparametric magnetic resonance (mpMRI) brain scans, each annotated with segmentation masks delineating tumorous regions. Each scan includes four modalities: Fluid Attenuated Inversion Recovery (FLAIR), native T1-weighted (T1), post-contrast T1-weighted (T1Gd), and T2-weighted (T2) images. Three recombined regions—the tumor core, the entire tumor, and the enhancing tumor—are used to quantify performance using 5-fold cross-validation.

**Spleen 3d Segmentation:** The Spleen 3D Segmentation dataset (Antonelli et al., 2022) focuses on segmenting spleens within portal-venous phase CT scans from patients undergoing chemotherapy treatment for liver metastases. The dataset consists of 61 volumetric CT scans, with 41 scans designated for training and the remaining 20 reserved for testing and evaluation. Each scan provides detailed 3D representations of abdominal anatomy, emphasizing the spleen and its surrounding structures during the portal-venous phase. The segmentation task involves accurately delineating the spleen, which is critical for assessing spleen-related conditions and treatment responses in oncology patients.

We used Dice loss and weighted ADAM optimizer for training. Dice similarity coefficient was used for quantitative evaluations. Our experiments used the PyTorch with MONAI (Cardoso et al., 2022) framework for model implementation. All experiments were conducted on a single NVIDIA A100 GPU (40GB).

### 3.2. Results

Figure 2 illustrates the relationship between the peak memory consumption during training and segmentation accuracy, measured by the Average Dice Score, across various models. It

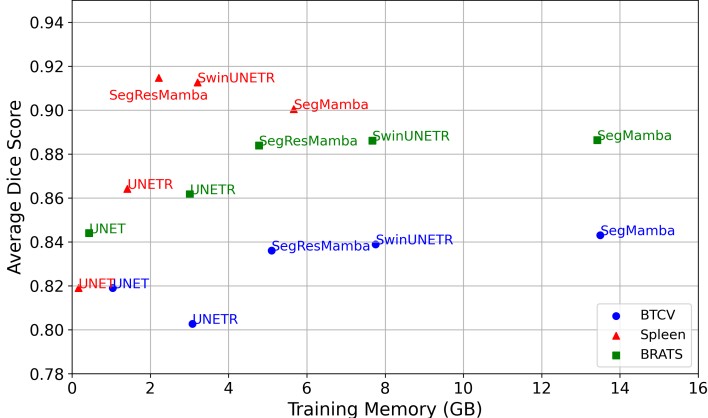

Figure 2: Average Dice Scores for BTCV, Spleen, and BraTS2021 datasets plotted against training memory (in GB) for different models using image size $128 \times 128 \times 128$ for BTCV and BRATS dataset and $96 \times 96 \times 96$ for Spleen dataset with batch size 1.

can be observed that our method uses comparatively less memory than other large models like Swin Unetr and SegMamba while still maintaining comparable performance.

Table 1 shows that SegResMamba, while having a smaller memory footprint and lower Multiply-Accumulate operations (MACs), achieves better or comparable performance on the BTCV dataset comparing with other more memory-intensive and computationally expensive models. SegResMamba outperforms nnFormer (Zhou et al., 2021), 3D UX-Net (Lee et al., 2022), and nnUNet (Isensee et al., 2021; Shaker et al., 2024) on computation cost, Inference Time as well as the segmentation performance.

| Model | MACs | Inference Time (sec) | Avg Dice |
|---|---|---|---|
| UNETR | 196.03G | 0.0531 | 0.8027 |
| SegMamba | 1554.86G | 0.1693 | 0.8430 |
| UNET | 60.10G | 0.0273 | 0.8192 |
| SwinUnetr | 784.46G | 0.1343 | 0.8389 |
| nnFormer | 648.10G | 0.0958 | 0.8239 |
| nnUNet | 1067.97G | 0.1668 | 0.8316 |
| 3D UX-Net | 1498.66G | 0.1338 | 0.8326 |
| SegResMamba | 336.45G | 0.0841 | 0.8361 |

Table 1: Average Dice scores of models on the BTCV dataset.

Brain tumor segmentation performances of different SOTA models are shown in Table 2. SegResMamba achieves a competitive mean Dice score of 0.8839, which is comparable to models like SwinUNETR (0.8861) and SegMamba (0.8863). Despite this, SegRes-Mamba operates with significantly lower MACs, making it a more computationally efficient

model. This lower computational cost, combined with its strong performance, highlights SegResMamba as an excellent choice for scenarios requiring a balance between accuracy and resource efficiency.

| Model | MACs | Mean Dice | Dice TC | Dice WT | Dice ET |
|---|---|---|---|---|---|
| UNETR | 203.29G | 0.8617 | 0.8653 | 0.8708 | 0.8490 |
| SegMamba | **1575.13G** | 0.8863 | 0.8943 | 0.8962 | 0.8685 |
| UNET | 30.13G | 0.8444 | 0.8435 | 0.8637 | 0.8260 |
| SwinUnetr | 792.08G | 0.8861 | 0.8907 | 0.8970 | 0.8707 |
| SegResMamba | 340.52G | 0.8839 | 0.8953 | 0.8958 | 0.8605 |

Table 2: Mean dice scores of different models on BraTS21 dataset for 5-fold cross-validation. Dice TC, Dice WT, and Dice ET represent the Dice scores for Tumor Core, Whole Tumor, and Enhancing Tumor, respectively.

| Model | MACs | Avg Dice |
|---|---|---|
| UNETR | 82.52G | 0.8642 |
| SegMamba | 655.32G | 0.9004 |
| UNET | 11.53G | 0.8195 |
| SwinUnetr | 328.68G | 0.9126 |
| SegResMamba | 137.84G | 0.9147 |

Table 3: Average Dice scores of models on the Spleen dataset.

In the spleen segmentation task, as shown in Table 3, the SegResMamba network achieved the highest average Dice score of **0.9147**, outperforming UNETR (0.8642), UNET (0.8195), and SwinUNETR (0.9126). This highlights SegResMamba's superior performance compared to transformer-based models on a small dataset.

| Exp | Model | Avg Dice (BTCV) |
|---|---|---|
| 1 | SegMamba Encoder + ResNet-based Decoder | 0.8164 |
| 2 | Exp. 1 + Convolution Mamba Mixed Block | 0.8279 |
| 3 | Exp. 2 + Additional Conv before downsampling | 0.8361 |

Table 4: Average Dice scores of different setups on the BTCV dataset.

To investigate the contribution of various components in our model, we conducted an ablation study on the BTCV dataset, with results shown in Table 4. In the first experiment, a Mamba encoder proposed by SegMamba(Xing et al., 2024) was paired with a lightweight ResNet-based decoder. This helps us to reduce computational complexity and memory efficiency. Next, we replaced the TSMamba block (Global Spatial Context (GSC) and ToM) used in the SegMamba encoder with our convolution mamba mixed block to enhance feature extraction of the encoder. This modification leverages both local representation through convolution and global representation via the mamba layer and improves the segmentation performance by 1.15%. Finally, we added a convolutional layer before the downsampling operation to preserve essential features. When combined with the improvements from Ex-

periments 1 and 2, this experiment led to significant performance gains, increasing the Dice score on the BTCV dataset from 0.8164 to 0.8361. To investigate the effectiveness of the ToM layer, we performed an ablation study with and without the ToM layer, the results shown in Table 5. 1.27% improvement on BTCV dataset with ToM layer shows the effectiveness of the Mamba module.

| Model | Avg Dice (BTCV) |
|---|---|
| Ours without ToM | 0.8234 |
| Ours with ToM (SegResMamba) | 0.8361 |

Table 5: Average Dice scores of different setups with and without ToM on the BTCV dataset.

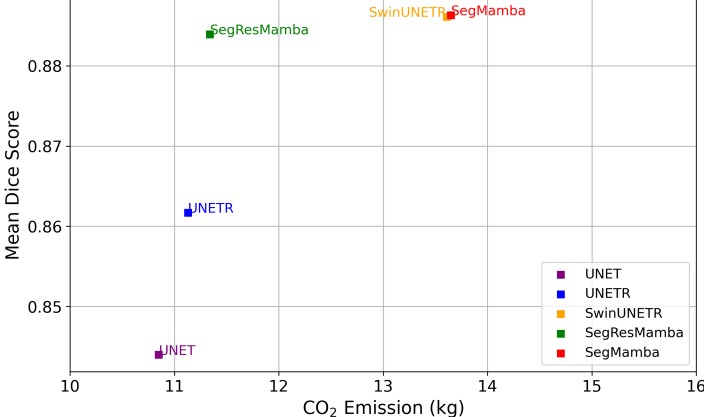

Figure 3: Mean dice score of BraTS dataset against $CO_2$ emission with 5-fold cross-validation settings for different models.

Furthermore, Figure 3 illustrates the relationship between $CO_2$ emission and segmentation accuracy for brain tumor segmentation with 5-fold cross-validation across various models. These estimations are based on training time and conducted using Amazon Web Services in region eu-central-1, which has a carbon efficiency of 0.61 kgCO₂eq/kWh. A cumulative training hours of computation was performed on hardware of type A100 PCIe 40GB (TDP of 250W). Estimations were conducted using the Machine Learning Impact calculator presented in (Lacoste et al., 2019). Among the high-performing models, SegResMamba demonstrates a notable advantage by achieving a balance between environmental efficiency and segmentation performance. Specifically, SegResMamba exhibits significantly lower $CO_2$ emissions compared to other high-performing models such as SwinUNETR and SegMamba, while maintaining a comparable dice score. Furthermore, when compared to UNET and UNETR, SegResMamba achieves superior segmentation accuracy without a substantial increase in $CO_2$ emissions, highlighting its efficiency and effectiveness.

## 4. Discussion

The experimental results demonstrate that SegResMamba is a robust and efficient model for 3D medical image segmentation tasks. It consistently delivers competitive performance across datasets while significantly reducing memory consumption and computational costs compared to state-of-the-art models like SwinUNETR and SegMamba. The model's design prioritizes memory efficiency without compromising segmentation accuracy. The reduced training memory requirements make this model an excellent choice for training and deployment on less resource-intensive hardware.

In terms of computational complexity, SegResMamba requires only 340.52 GMACs for the BraTS21 dataset (Table 2), a significant improvement over SegMamba (1575.13 GMACs) and SwinUNETR (792.08 GMACs). Despite its lightweight design, SegResMamba maintains a competitive mean Dice score of 0.8839, only 0.24% and 0.22% less than SegMamba and SwinUNETR respectively. This demonstrates the model's ability to achieve high segmentation accuracy while remaining computationally efficient which makes it more suitable to be deployed in energy-sensitive situations.

SegResMamba's performance across datasets further highlights its versatility. On the BTCV dataset, the model achieves Dice scores comparable to memory-intensive counterparts like SegMamba and SwinUNETR (Table 1), while attaining the highest Dice score of 0.9147 on the spleen segmentation task (Table 3). These results emphasize its effectiveness in addressing diverse segmentation challenges.

Environmental efficiency is another key aspect of the proposed model. SegResMamba demonstrates significantly lower $CO_2$ emissions compared to other SOTA models during training due to reduced memory and computational requirements. This aligns with sustainable AI practices, promoting the development of energy-efficient models that minimize environmental impact without compromising performance.

While SegResMamba demonstrates substantial advantages, there are a few limitations to consider. First, its segmentation performance, although competitive, is marginally lower than other high-performing models like SwinUNETR and SegMamba, as observed in the BraTS and BTCV datasets. This slight trade-off may be a consideration for applications where peak accuracy is critical. Another limitation is that the training and evaluation were performed on three datasets with well-defined segmentation tasks; performance on more challenging, larger, or less-structured datasets remains to be explored.

## 5. Conclusion

SegResMamba marks a significant advancement in 3D medical image segmentation, balancing efficiency and performance by combining Mamba's global context modeling with convolutional layers for local feature extraction. Its reduced memory overhead, along with improved computational and training efficiency, makes it well-suited for real-world clinical applications, delivering excellent results while remaining resource-efficient. Future work will focus on exploring new training strategies and data augmentation to further enhance segmentation accuracy and generalization across various datasets.

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

## Appendix A. Additional Implementation Details

### A.1. Brain Tumor Segmentation

BraTS2021 dataset (Baid et al., 2021) was used for brain tumor segmentation to compare the performance across multiple folds for the SOTA models. We trained 5-fold cross-validation for 200 epochs utilizing strategies like learning rate scheduling with CosineAnnealing, Adam optimizer with weight decay of 1e-5, and gradient scaling. We used dice metric and dice loss as metric and the loss function. Various dataset transforms like foreground cropping, random spatial cropping, random flip with probability 0.5 in each direction, and random intensity scaling were used.

### A.2. Multi-organ Segmentation

We conducted experiments on the BTCV dataset for multi-organ segmentation (Landman et al., 2015). The training process ran for 25,000 steps. We utilized the Adam optimizer with a learning rate of 1e-4 for our experiments. Our data transformations included scaling intensity range, orientation adjustment (Orientationd), foreground cropping (CropForegroundd), and spacing adjustment (Spacingd). To optimize the model's performance, we employed DiceLoss as the loss function and evaluated using the dice metric for validation.

### A.3. Spleen Segmentation

For the spleen segmentation task, we used the spleen 3D segmentation dataset (Antonelli et al., 2022) and we limited training to 100 epochs. Extending the training to larger epoch numbers, such as 200, results in overfitting due to the relatively small size of the dataset compared to larger datasets like BraTS. Following a similar approach to the aforementioned tasks we used Adam optimizer with a learning rate of 1e-4. Transformations like scaling intensity range, normalizing the orientations of images, foreground cropping, and spacing adjustment were used. DiceLoss was used as the loss function and dice metric as the metric for validation.

## Appendix B. Training Time and $CO_2$ Footprint

| Model | Epoch Time (in sec) | Total Time (in hours) | 5-Fold Time (in hours) | $CO_2$ Emissions (kg) | | |
|---|---|---|---|---|---|---|
| | | | | Azure | Google | Amazon |
| UNETR | 262.83 | 14.60 | 73.01 | 10.40 | 11.32 | 11.13 |
| Segmamba | 321.50 | 17.86 | 89.31 | **12.73** | **13.84** | **13.62** |
| UNET | 255.80 | 14.21 | 71.06 | 10.13 | 11.01 | 10.84 |
| SwinUNETR | 321.39 | 17.85 | 89.28 | 12.72 | 13.84 | 13.61 |
| Segresmamba | 267.83 | 14.88 | 74.40 | 10.60 | 11.53 | 11.35 |

Table 6: Comparison of models in terms of training time, and $CO_2$ emissions across different cloud providers for training of 5-fold cross-validation using BraTS dataset

A detailed comparison of $CO_2$ emissions across different cloud providers, including Amazon Web Services, Google Cloud, and Azure, for 5-fold training of the BraTS dataset, is presented in Table 6. It is important to note that these values represent only the emissions from 5-fold training; incorporating hyperparameter optimization would result in significantly higher $CO_2$ emissions. These estimations were conducted using the Machine Learning Impact calculator presented in (Lacoste et al., 2019).

## Appendix C. Models Configuration

Table 7 presents the model configurations, including the number of parameters (in millions) and FLOPs for each method for the BTCV dataset. Our proposed model, SegResMamba, has 188.42G FLOPs, striking a balance between computational efficiency and performance compared to more complex architectures like SwinUNETR, SegMamba and 3D UX-Net.

| Model | Num of Params (in million) | FLOPs |
|---|---|---|
| UNETR | 93.01 | 177.44G |
| SegMamba | 67.36 | 1443.96G |
| UNET | 4.89 | 25.93G |
| SwinUnetr | 62.19 | 767.23G |
| nnFormer | 149.32 | 426.74G |
| nnUNeT | 31.19 | 480.06G |
| 3D UX-Net | 53.01 | 1373.75G |
| SegResMamba | 119.98 | 188.42G |

Table 7: Num of params of methods with FLOPs count

## Appendix D. Memory Efficiency

A comparison of training memory of different models is shown in Table 8 (using image size $128 \times 128 \times 128$ for BTCV and BraTS dataset and $96 \times 96 \times 96$ for Spleen dataset with batch size 1).

| Model | BTCV (GB) | Spleen (GB) | BRATS (GB) |
|---|---|---|---|
| **UNETR** | 3.08 | 0.14 | 3.02 |
| **SegMamba** | 13.51 | 5.68 | 13.44 |
| **UNET** | 1.42 | 0.48 | 1.13 |
| **SwinUNETR** | 7.77 | 3.21 | 7.68 |
| **SegResMamba** | 5.10 | 2.22 | 4.78 |

Table 8: Training memory (in GB) for different models on BTCV, Spleen, and BraTS datasets.

