# OpenReview forum: "SegResMamba: An Efficient Architecture for 3D Medical Image Segmentation"
_MIDL.io/2025/Conference — MIDL 2025 Poster_

### Official Review · Reviewer_MY4P · 2025-02-16

**Confidence:** 2
**Preliminary Rating:** 2
**Recommendation:** Poster
**Final Rating:** 4

**Summary:**

The paper proposes to address some computational and memory issues of segmentation models beyond CNNs, ie. transformers and Mamba (SSMs). In particular the authors propose a few technical choices for improving efficiencies while maintaining, including a tri-oriented Mamba sequence and convolution mamba mixed blocks. The experiments demonstrate certain gains against other Mamba or Transformer networks but do not broadly compare to SOTA UNets e.g. the nnUNET.

**Strengths:**

- practical improvements for memory reduction to save computation time and energy
- detailed comparisons of memory usage on two datasets against SOTA (except nnUNet)
- ablation of three different settings for one dataset

**Weaknesses:**

- the paper seems hastily written and does not clearly convey the key concepts of the proposed work
- the inter-slice dependency of the ToM module is uni-directional in z introducing a strong bias regarding the orientation of 3D volumes
- the technical /implementation details are somewhat unfitting for the method section, making it seem like network engineering rather than a conceptually new approach. It remains unclear where exactly the memory gains are reached
- there are no time/speed measurements compared to CNN-UNets but the MACs of SegResMamba still are 5-10x higher, hence a much lower efficacy is expected.

**Detailed Comments:**

Quite a few details remain unclear and should be better motivated / explained in a revised version
1) how does the network compare to the nnUNet and how is the compared UNet parameterised?
2) why is there a zig-zag order for the two axial dimensions but simple unconnected rays for inter-slice interactions in ToM?
3) what is the exact difference of the ConvMambaMixBlock compared to prior work, in particular I am missing any discussion on Vision Mamba (https://arxiv.org/abs/2401.09417) and/or VMamba (https://arxiv.org/abs/2401.10166) . Quoting from their abstracts: "Vim is 2.8× faster than DeiT and saves 86.8% GPU memory when performing batch inference to extract features on images with a resolution of 1248×1248" and  "By traversing along four scanning routes, SS2D bridges the gap between the ordered nature of 1D selective scan and the non-sequential structure of 2D vision data, which facilitates the collection of contextual information from various sources and perspectives. Based on the VSS blocks, we develop a family of VMamba architectures and accelerate them through a succession of architectural and implementation enhancements"
4) since the replacement of the Global Spatial Context of SegMamba comes along other architectural changes it would be interesting to understand how a related CNN (proposed architecture just skipping ToM) would work?
5) The CO2 analysis and MACs seem to give unrealistic results, specifically for the UNet. A UNet with 3x3x3 convolutions and sufficient channel width typically runs at near optimum efficiency (ie lower MACs = better speed). Table1+3 show SegResMamba to have 10x more MACs than the UNet, but the CO2 measurement in Fig. 3 only shows 5-10% lower runtimes (this seems strange) - to reiterate: the UNet choice seems not ideal because the number of MACs is extremely low - for reference an nnUNet usually has >>500 GMACs.
6) A more detailed SOTA comparison could be beneficial e.g. https://github.com/MIC-DKFZ/nnUNet/blob/master/documentation/resenc_presets.md lists quite a few recent advances on BTCV and BRATS (reaching often 3-4% points higher Dice scores than reported here), http://medicaldecathlon.com/results/ lists the results on Spleen with top scores of 96-97% Dice (much better than any result in this paper)

**Justification Of The Final Rating:**

While I highly appreciate the authors response, while I initially planned to stick to my original rating the final added ablation persuaded me to upgrade my score. The paper has improved during rebuttal and the authors mentioned and performed many interesting ablation studies that are helpful. Hence, overall this is a good mix of novelty and experimental validation that would be of interest for the MIDL community. I still believe that further experiments are needed but the authors do demonstrate the positive effect of the proposed 3D Mamba block. I would recommend to improve the quality of the figures (maybe add some distribution plots instead of point averages and increase font size).

**Justification Of The Preliminary Rating:**

While not uninteresting I feel the paper is not yet ready for publication as it misses a clearer focus and better experimental validation. I am inclined to increase my score if there is sufficient reasoning for various choices and a more detailed discussion compared to related work.

**Questions To Address In The Rebuttal:**

In particular the questions regarding the missing ablations, comparisons and better differentiation from prior work would be relevant. In addition the authors should communicate the details of their UNet architecture (and why they chose not to use the nnUNet) along with some actual time measurements (inference/training), there are some numbers in the appendix but as mentioned before I cannot make sense of them.

**Special Issue:**

No

---

> ### Author Response · Authors · 2025-03-08
> **Official Comment by Authors**
>
> 1.	how does the network compare to the nnUNet and how is the compared UNet parameterised?
> -	We have added comparison with nnUNet, nnFormer, and 3D UX-Net in the manuscript (in Table 1). Also, for UNET we used the MONAI framework with the default configurations. We mentioned in the manuscript “Our experiments used the PyTorch framework with MONAI for model implementation.”
> 2.	why is there a zig-zag order for the two axial dimensions but simple unconnected rays for inter-slice interactions in ToM?
> -	We thank the reviewer for spotting this. We have modified the Figure to make it clear. Please note the inter-slice dashed lines we have added in the inter-slice feature extraction figure of ToM (the last figure of ToM).
> 3.	what is the exact difference of the ConvMambaMixBlock compared to prior work, in particular I am missing any discussion on Vision Mamba (https://arxiv.org/abs/2401.09417) and/or VMamba (https://arxiv.org/abs/2401.10166) . Quoting from their abstracts: "Vim is 2.8× faster than DeiT and saves 86.8% GPU memory when performing batch inference to extract features on images with a resolution of 1248×1248" and "By traversing along four scanning routes, SS2D bridges the gap between the ordered nature of 1D selective scan and the non-sequential structure of 2D vision data, which facilitates the collection of contextual information from various sources and perspectives. Based on the VSS blocks, we develop a family of VMamba architectures and accelerate them through a succession of architectural and implementation enhancements"
>
> -	We have modified the methodology part to provide more details about our design and Conv Mamba Mixed Block. Vision Mamba and VMamba were designed for 2D natural images. We introduce Conv mamba mixed block which uses Tri-orientated Mamba (ToM) blocks and convolution for feature extraction. Specifically, the VSS Block of VMamba used 3X3 depth convolution, but we used multiple large kernel 3D convolution to improve the local feature extraction instead. Vision Mambe used 1D convolution in their encoder to do convolution on the sequence of tokens, but we used 3D convolution to extract 3D spatial features.
> 4.	since the replacement of the Global Spatial Context of SegMamba comes along other architectural changes it would be interesting to understand how a related CNN (proposed architecture just skipping ToM) would work?
> -	Basically, it would be very similar to the compared UNET. We would like to include more of this type of ablation study in our future work.
> 5.	The CO2 analysis and MACs seem to give unrealistic results, specifically for the UNet. A UNet with 3x3x3 convolutions and sufficient channel width typically runs at near optimum efficiency (ie lower MACs = better speed). Table1+3 show SegResMamba to have 10x more MACs than the UNet, but the CO2 measurement in Fig. 3 only shows 5-10% lower runtimes (this seems strange) - to reiterate: the UNet choice seems not ideal because the number of MACs is extremely low - for reference an nnUNet usually has >>500 GMACs.
> -	The CO2 analysis was performed using an online Machine Learning Impact Calculator (https://mlco2.github.io/impact/) based on training time, as detailed in Appendix B. While the GMAC count for UNet is significantly lower, its training time on a single NVIDIA A100 GPUs(40GB) was 14.21 hours, compared to 14.88 hours for SegResMamba. This explains why the difference in CO2 emissions appears smaller than expected. We have clarified this in the main manuscript: “These estimations are based on training time and conducted using Amazon ….” Meanwhile, for UNET we used the Monai implementation and the default configuration. We also added nnUNet in our comparison in Table 1.
> 6.	A more detailed SOTA comparison could be beneficial e.g. https://github.com/MIC-DKFZ/nnUNet/blob/master/documentation/resenc_presets.md lists quite a few recent advances on BTCV and BRATS (reaching often 3-4% points higher Dice scores than reported here), http://medicaldecathlon.com/results/ lists the results on Spleen with top scores of 96-97% Dice (much better than any result in this paper)
> -	We have added additional comparisons with nnUNet, nnFormer, and 3D UX-Net on the BTCV dataset. To ensure a fair evaluation, we ran all our methods using the MONAI pipeline for the same number of epochs, selecting model checkpoints that achieved the best results on the validation datasets for testing. Some methods referenced in the provided link may have different experimental configurations, such as a higher number of epochs, ensembling, or train and test-time augmentation. However, our objective is not solely to maximize the Dice score but to strike a balance between computational cost and performance.

---

> > ### Comment · Reviewer_MY4P · 2025-03-13
> >
> > I would like to thank the authors for their rebuttal and updated manuscript. Unfortunately, the PDF contains no mark-up making it hard to track changes. Did the authors mention the fact that the additional nnUNet (etc) results are not following the default configurations chosen by the respective authors for their challenge entries? This is relevant information for readers to better assess the presented results.
> >
> > Regarding the missing ablation of the modified MONAI UNet before and after insertion of TOM (in line 5+9 in pseudo code). I still think this is crucial: as we can see from the parameter count the UNET is not competitively parameterised and it remains unclear whether the proposed Mamba step (TOM) is really adding any value.

---

> > ### Author Response · Authors · 2025-03-13
> > **Official Comment by Authors**
> >
> > We sincerely appreciate the reviewer’s valuable feedback and suggestions.
> >
> > For model configurations, we followed the default settings from MONAI (e.g., UNETR, SwinUNETR) or the respective codebases of the methods (e.g., nnUNet, nnFormer). However, for experimental configurations, we adopted a standardized approach to ensure fair evaluation. All methods were trained for the same number of epochs, selecting model checkpoints that achieved the best results on the validation datasets for testing.
> > As we cannot modify the rebuttal PDF now, we will clarify this in our revised final manuscript.
> >
> > Regarding the ablation study, we acknowledge the reviewer’s concern. To address this, we would like to conduct an additional ablation study by removing the ToM layer. We will include the result in the final manuscript.

---

> ### Author Response · Authors · 2025-03-15
> **Official Comment by Authors | Ablation Study**
>
> We have conducted the additional ablation study suggested by the reviewer. Please find the result below:
>
> | Model    | Average Dice (BTCV) |
> | -------- | ------- |
> | SegResMamba without ToM  | 0.8234    |
> | SegResMamba with ToM | **0.8361**     |
>
> This result shows the positive effect of the Mamba block in our architecture.
> As mentioned earlier, we will include this result in our final manuscript.

---

> > ### Comment · Reviewer_MY4P · 2025-03-15
> >
> > Thanks! This now convinced me to increase my score. Please note that I would not call the first line SegResMamba anymore or at least put the Mamba in brackets.

---

### Official Review · Reviewer_29W9 · 2025-02-21

**Confidence:** 5
**Preliminary Rating:** 3
**Recommendation:** Poster
**Final Rating:** 3

**Summary:**

This paper introduces a memory-efficient and computationally light approach for 3D medical image segmentation, built upon a hybrid design that fuses convolutional layers with Structured State Space Models (SSMs), specifically Mamba-based blocks. By leveraging a Tri-oriented Mamba (ToM) mechanism and convolution mamba mixed blocks, the architecture captures both local and long-range dependencies in volumetric scans. It manages to substantially reduce memory consumption and computational load while preserving robust performance on multiple datasets (e.g., BraTS, BTCV, and Spleen).

**Strengths:**

1.	The balanced integration of convolutional layers and Mamba blocks, which clearly addresses the local and global feature extraction issues in 3D volumes.
2.	Systematically evaluating on multiple datasets, including small-scale (Spleen) and more comprehensive ones (BTCV, BraTS), the authors convincingly show that SegResMamba maintains near state-of-the-art performance with significantly reduced GPU memory usage and overall computational cost.

**Weaknesses:**

1.	The paper does not compare its results with some state-of-the-art (SOTA) baselines, such as nnFormer or UNETR++.
2.	The paper should also report parameters, FLOPs, and inference speed to demonstrate the model’s effectiveness in reducing computational complexity, memory usage, and training time.
3.	Although the authors mention the model’s environmental advantages, they do not provide details on fine-grained hyperparameter tuning or resource usage (e.g., training on multi-GPU setups or CPU-only scenarios).

**Detailed Comments:**

1.	See the weakness
2.	Additional ablation on the number of Mamba blocks or the tri-oriented Mamba design (e.g., comparing forward-only vs. tri-oriented usage) could confirm the necessity of each orientation.
3.	Since Mamba-based blocks are relatively new, sharing code snippets or best practices would accelerate adoption.

**Justification Of The Final Rating:**

The authors do compare their method to some state-of-the-art (SOTA) approaches, but they do not include all of them (e.g., UNETR++). My main concern is that the proposed method does not offer significant improvements in either efficiency or effectiveness. Therefore, I maintain my borderline rating.

**Justification Of The Preliminary Rating:**

This paper introduces a memory-efficient and computationally light approach for 3D medical image segmentation, built upon a hybrid design that fuses convolutional layers with Structured State Space Models (SSMs), specifically Mamba-based blocks.  However, due to the weakness mentioned, I recommend Broadline.

**Questions To Address In The Rebuttal:**

1.	Could you provide more details on what minor architectural or hyperparameter changes might close the small accuracy gap with SwinUNETR while preserving SegResMamba’s efficiency?
2.	Please include comparisons with UNETR++ and nnFormer.
3.	Report the parameters, FLOPs, and inference speed of the models.

**Special Issue:**

No

---

> ### Author Response · Authors · 2025-03-08
> **Official Comment by Authors**
>
> 1.	Could you provide more details on what minor architectural or hyperparameter changes might close the small accuracy gap with SwinUNETR while preserving SegResMamba’s efficiency?
>
> -	We utilized a powerful encoder comprising of Tri-oriented Mamba and convolutional layers to effectively capture both local and global features. We then use a lightweight decoder to generate segmentation mask. A simplified decoder design minimizes redundant computations, further optimizing memory efficiency. In this way, the computational cost of the model is reduced and the responsibility of maintaining the model's performance is on the uniquely designed, powerful yet efficient encoder. The decoder achieved light-weight by using sum instead of concatenation when fuse the feature maps from the decoder and encoder, and replacing heavy CNN blocks with an MLP layer on the skip connection. We have updated this information in the methodology part of the manuscript.
> -	Further improving performance would likely require additional architectural refinements, such as increasing the model's capacity or incorporating additional feature refinement mechanisms. However, these enhancements would come at the cost of higher memory consumption and computation cost, highlighting the inherent tradeoff between performance and efficiency. Our focus remains on maintaining a balance that ensures strong segmentation performance while preserving computational efficiency, making the model more practical for resource-constrained settings.
>
> 2.	Please include comparisons with UNETR++ and nnFormer.
> -	We thank the reviewer for the suggestion. We have included nnUNet, nnFormer, and 3D UX-Net in our comparison for the BTCV dataset (in Table1) within the rebuttal period. We would like to consider more comparisons in the future work.
>
> 3.	Report the parameters, FLOPs, and inference speed of the models.
> -	The number of parameters and FLOPs are added to Appendix C. We also added inference speed for the BTCV dataset in Table 1.
>
> 4.	Although the authors mention the model’s environmental advantages, they do not provide details on fine-grained hyperparameter tuning or resource usage (e.g., training on multi-GPU setups or CPU-only scenarios).
> -	We appreciate the reviewer’s suggestion. All experiments were conducted on a single NVIDIA A100 GPU (40GB), and we have added this information to the "Dataset and Implementation Details" subsection. We also have provided hyperparameter details, including learning rates, and weight decay in Appendix A.

---

### Official Review · Reviewer_UGvW · 2025-02-24

**Confidence:** 5
**Preliminary Rating:** 2
**Final Rating:** 2

**Summary:**

This paper introduces SegResMamba, a 3D medical image segmentation model that leverages Structured State Space Models (SSMs) to address the computational and memory challenges of Transformer-based architectures. While Transformers have demonstrated strong performance in capturing long-range dependencies, their application to 3D medical imaging is mainly hindered by high training costs and memory demands. SegResMamba is designed to reduce computational complexity, memory usage, and training time while maintaining competitive segmentation accuracy. The model reportedly uses less than half the memory of state-of-the-art (SOTA) architectures, making it a more scalable and environmentally sustainable alternative. However, stronger comparisons with SOTA 3D segmentation models and a deeper analysis of efficiency-performance trade-offs are needed to solidify its impact.

**Strengths:**

1. This paper introduces an efficient architecture for 3D medical image segmentation, which is interesting and may have practical implications for the medical AI community.
2. The paper is well-structured and clearly written, making it easy to follow and understand.

**Weaknesses:**

1. Limited contribution:
(1) The Tri-orientated Mamba structure has already been introduced in SegMamba, and the paper does not clearly differentiate SegResMamba from U-Mamba and SegMamba. A clear comparison and discussion highlighting SegResMamba’s unique contributions over these existing models is necessary.
(2) Although SegResMamba reduces MACs, Table 1 and Table 2 show that it lowers Dice scores for ET segmentation on BraTS (-0.8%) and BTCV (-0.69%). In medical applications where accuracy is paramount, this trade-off raises concerns about SegResMamba’s clinical applicability and generalization.
2. Missing essential baseline comparisons:
(1) The paper lacks comparisons with state-of-the-art models, particularly nnU-Net [1], which remains the gold standard in medical image segmentation.
(2) Other strong baseline models, such as 3D UX-Net [2], which utilizes large-kernel depthwise convolutions, and Mamba-based models like U-Mamba [3] and SwinUMamba [4], should be included to provide a comprehensive evaluation of SegResMamba’s performance in a broader context.
3. Lack of validation on larger and more diverse datasets: The model’s effectiveness should be further validated on larger and more diverse datasets, such as Totalsegmentator and the latest BraTS 2024, to demonstrate robustness and generalizability across different organs and pathologies.

References:
[1] Isensee F, Jaeger P F, Kohl S A A, et al. nnU-Net: a self-configuring method for deep learning-based biomedical image segmentation[J]. Nature methods, 2021, 18(2): 203-211.
[2] Lee, H.H., Bao, S., Huo, Y. and Landman, B.A., 2022. 3d ux-net: A large kernel volumetric convnet modernizing hierarchical transformer for medical image segmentation. arXiv preprint arXiv:2209.15076.
[3] Jun Ma, Feifei Li, and Bo Wang. U-mamba: Enhancing long-range dependency for biomedical image segmentation. arXiv preprint arXiv:2401.04722, 2024.
[4] Liu, J., Yang, H., Zhou, H.Y., Xi, Y., Yu, L., Li, C., Liang, Y., Shi, G., Yu, Y., Zhang, S. and Zheng, H., 2024, October. Swin-umamba: Mamba-based unet with imagenet-based pretraining. In International Conference on Medical Image Computing and Computer-Assisted Intervention (pp. 615-625). Cham: Springer Nature Switzerland.

**Detailed Comments:**

All comments are mentioned in Weaknesses

**Justification Of The Final Rating:**

The authors have indeed added comparisons between their method and some advanced baselines in the rebuttal. However, their evaluation is primarily focused on the relatively small BTCV dataset, while comparisons with nnU-Net, nnFormer, and 3D UX-Net on larger-scale datasets, such as BraTS, are missing. Moreover, the paper lacks a comparison with the important baseline U-Mamba. Given these limitations, it remains difficult to be convinced of the proposed model’s contribution to the field. Therefore, I will maintain my original score.

**Justification Of The Preliminary Rating:**

The paper presents SegResMamba, but its contribution is limited, and the differences from U-Mamba&SegMamba are not clearly justified. Additionally, performance trade-offs in medical segmentation raise concerns about its clinical applicability, and the lack of key baseline comparisons (e.g., nnU-Net, 3D UX-Net, U-Mamba) further weakens the evaluation, leading to a Weak Reject rating.

**Questions To Address In The Rebuttal:**

please see Weaknesses

---

> ### Author Response · Authors · 2025-03-08
> **Official Comment by Authors**
>
> 1.	Limited contribution: (1) The Tri-orientated Mamba structure has already been introduced in SegMamba, and the paper does not clearly differentiate SegResMamba from U-Mamba and SegMamba. A clear comparison and discussion highlighting SegResMamba’s unique contributions over these existing models is necessary. (2) Although SegResMamba reduces MACs, Table 1 and Table 2 show that it lowers Dice scores for ET segmentation on BraTS (-0.8%) and BTCV (-0.69%). In medical applications where accuracy is paramount, this trade-off raises concerns about SegResMamba’s clinical applicability and generalization.
> -	(1) We proposed a convolution mamba mixed block to replace TSMamba block from SegMamba for better feature extraction in the encoder. From the ablation study (Table 4), it can be seen that when we used an encoder with Conv Mamba Mixed Block, the segmentation performance improved. We also have a very lightweight decoder compared to SegMamba, which helps us to reduce computational cost and memory usage. We have discussed this difference in the discussion section and we also revised our methodology for better clarity.
> -	(2) We agree with the reviewer's point. There is always a tradeoff between performance and computational cost. While our performance may be slightly lower than some SOTA methods, it remains comparable. However, we want to emphasize that achieving this level of performance with significantly reduced computational cost is a crucial advantage, particularly for resource-constrained clinical applications.
> 2.	Missing essential baseline comparisons: (1) The paper lacks comparisons with state-of-the-art models, particularly nnU-Net [1], which remains the gold standard in medical image segmentation. (2) Other strong baseline models, such as 3D UX-Net [2], which utilizes large-kernel depthwise convolutions, and Mamba-based models like U-Mamba [3] and SwinUMamba [4], should be included to provide a comprehensive evaluation of SegResMamba’s performance in a broader context.
> -	We have added new experiments comparing SegResMamba with nnU-Net, nnFormer, and 3D UX-Net on the BTCV dataset (in Table 1) as suggested by the reviewers within the limited time for rebuttal.  We would like to include comparison with more methods like U-Mamba and SwinUMamba on more datasets in our future work.
> 3.	Lack of validation on larger and more diverse datasets: The model’s effectiveness should be further validated on larger and more diverse datasets, such as Totalsegmentator and the latest BraTS 2024, to demonstrate robustness and generalizability across different organs and pathologies.
> -	We thank the reviewer for the suggestion. We have validated the performance of SegResMamba on three datasets across different organs. The differences between the BraTS 2021 and BraTS 2023 (latest) datasets are minimal. We agree that training on additional large-scale datasets like TotalSegmentator would further strengthen our findings, but it is challenging due to the limit amount of time for rebuttal. We acknowledge this as a limitation and have highlighted it in the discussion section. We would like to include this in our future work.

---

### Official Review · Reviewer_4y9X · 2025-02-27

**Confidence:** 5
**Preliminary Rating:** 4
**Final Rating:** 4

**Summary:**

SegResMamba represents a major advancement in 3D medical image segmentation by integrating Mamba’s global context modeling with convolutional layers for efficient local feature extraction. This hybrid architecture enhances segmentation accuracy while significantly reducing memory consumption and computational overhead, making it a practical solution for real-world clinical applications. Key experiments demonstrate that SegResMamba achieves high segmentation performance with improved training efficiency compared to existing models, suggesting its potential for broader adoption in medical imaging tasks.

**Strengths:**

The major strength of the paper lies in the SegResMamba model's ability to drastically reduce memory consumption, requiring less than half the memory during training compared to other state-of-the-art (SOTA) architectures. Despite its lower resource demands, it maintains competitive performance, making it highly efficient and well-suited for real-world clinical applications where computational constraints are a key concern.

**Weaknesses:**

A critical concern of this paper is the justification for integrating Mamba’s global context modeling with convolutional layers for efficient local feature extraction. Additionally, the paper should provide a clearer explanation of the specific factors contributing to reduced memory overhead, which enhances computational efficiency without compromising segmentation accuracy. While the experimental results demonstrate the effectiveness of the proposed approach, a more rigorous mathematical justification would strengthen the claims and provide deeper insights into the model’s efficiency.

**Detailed Comments:**

The Methodology section primarily focuses on the technical implementation of the network structure, which could be more appropriately placed in the supplementary materials as implementation details. Instead, the authors should clearly articulate the motivation behind integrating Mamba with convolutional layers—what specific challenges or limitations does this combination address? Additionally, they should explain how this fusion contributes to the lightweight nature of SegResMamba. Without such justification, the integration may appear as an arbitrary modification rather than a well-founded design choice.

**Justification Of The Final Rating:**

The paper is not clearly written enough to effectively highlight the contribution of the proposed approach. Although the author claims to have addressed certain weaknesses in the revised version, I did not find any noticeable modifications.

**Justification Of The Preliminary Rating:**

The SegResMamba model introduces an innovative hybrid approach, combining Mamba’s global context modeling with convolutional layers to enhance 3D medical image segmentation. A major strength of the paper is its significant reduction in memory overhead, requiring less than half the memory of state-of-the-art (SOTA) architectures during training while maintaining competitive segmentation accuracy.
However, the paper could be further strengthened with clearer theoretical justifications for combining Mamba with convolutional layers. While the experimental results validate the effectiveness of the approach, a more rigorous mathematical or algorithmic explanation of the model’s efficiency gains would enhance credibility.

**Questions To Address In The Rebuttal:**

In their rebuttal, the authors should address the following key points:

1.	Justification for Combining Mamba with Convolutional Layers

a. How does Mamba’s global context modeling complement convolutional layers for local feature extraction?

b. Is there a theoretical or empirical basis demonstrating why this combination improves segmentation performance and efficiency?

2.	Explanation of Reduced Memory Overhead

a. What specific architectural contribute to SegResMamba’s significantly lower memory usage?

b. How does it compare in terms of FLOPs and parameter efficiency against SOTA models?

c. Can the authors provide a deeper mathematical or algorithmic explanation for this improvement?

3.	Clarification on the Methodology Section

a. The Methodology section currently reads as an implementation description rather than a conceptual justification.

b. The authors should highlight the design motivations before diving into technical details.

c. Could they move the lower-level implementation details to supplementary materials and focus more on the reasoning behind their choices?

---

> ### Author Response · Authors · 2025-03-08
> **Official Comment by Authors**
>
> 1.	Justification for Combining Mamba with Convolutional Layers
>
>    a. How does Mamba’s global context modeling complement convolutional layers for local feature extraction?
>
>      • Mamba efficiently captures long-range dependencies and global context, while convolutional layers excel at local feature extraction. Combining them balances global and local information, enhancing segmentation accuracy.
>
>
>     b. Is there a theoretical or empirical basis demonstrating why this combination improves segmentation performance and efficiency?
>
>      • Yes, both theoretical and empirical evidence supports this combination: Theoretically, Mamba’s linear-time complexity enables efficient global context modeling, while CNNs preserve local spatial details, ensuring precise segmentation. Empirically, our experiments show comparable performance despite having lower GMac and memory usage. Ablation studies confirm both components contribute to performance gains, and efficiency analysis highlights reduced computational cost compared to transformers, making the model suitable for medical imaging.
>
> 2.	Explanation of Reduced Memory Overhead
>
>    a.	What specific architectural contribute to SegResMamba’s significantly lower memory usage?
>
>    •  SegResMamba reduces memory usage through:
>     1.	Mamba’s Linear Complexity – Unlike transformers, Mamba processes sequences with linear complexity, lowering memory overhead.
>     2.	Lightweight Decoder – A simplified decoder design minimizes redundant computations, further optimizing memory efficiency.
>     We have modified the methodology section decoder part for more clarity.
>
>    b. How does it compare in terms of FLOPs and parameter efficiency against SOTA	 models?
>
>      •  We have added FLOPS and number of parameter details in Appendix C as suggested by the reviewer. In the main manuscript, we provide multiply-accumulate (MAC) operations, a key indicator of computational efficiency. We measured this using the ptflops library. MAC is a good metric for computational cost measurement as it more accurately reflects the dominant operation performed in deep learning, which is a multiplication followed by an addition, effectively capturing the core computation within a single unit, while FLOPs might count individual multiplications and additions separately.
>
>    c. Can the authors provide a deeper mathematical or algorithmic explanation for this improvement?
>
> 	• We have updated the method section specially the convolution mamba mixed block with an algorithmic view for better clarity.
>
> 3.	Clarification on the Methodology Section
> a. The Methodology section currently reads as an implementation description rather than a conceptual justification.
> b. The authors should highlight the design motivations before diving into technical details.
> c. Could they move the lower-level implementation details to supplementary materials and focus more on the reasoning behind their choices?
> - We thank the reviewer for the suggestion. We have modified our method section to provide details of our design motivations as suggested by the reviewer.

---

> ### Author Response · Authors · 2025-03-10
> **Discussion with Reviewer 4y9X**
>
> - We have modified the method section and mentioned why we designed our model like this clearly. (Methodology 1st paragraph, encoder 1st paragraph and decoder 1st paragraph)
> - We have also added num of params and FLOPS count as the reviewer requested.
> - We have provided an algorithm for our introduced convolution mamba mixed block as suggested by the reviewer.
> - We explained how memory and computational cost is reduced using lightweight decoder and mamba.
> - We have also added more comparisons with methods like nnUNet, nnFormer and 3D UX-Net.
> - Please let us know which part is not clear, so that we can work on that to improve our manuscript.

---

### Comment · Area_Chair_ycx7 · 2025-03-04
**rebuttal process**

Dear Authors,

I encourage you to actively participate in the rebuttal process, with a hard deadline on March 7 23:59 AoE.

You can leave official comments on OpenReview to address each reviewer's points separately, and upload a revised manuscript using the “rebuttal” function in OpenReview. The revision can include any additional details, experiments, or images that might be required in the paper within the page limit (max 9 excluding references, acknowledgements, and appendices), as well as other supporting documents in the rebuttal stage. Any changes must be highlighted in the revised manuscript.

Thank you

---

### Author Rebuttal · Authors · 2025-03-08

**Rebuttal:**

We sincerely thank all the Reviewers for their feedback. Specifically, we are grateful to Reviewer 4y9X, Reviewer MY4P, and Reviewer UGvW for acknowledging the efficiency and usefulness of the method in real-world clinical applications, and Reviewer 29W9 for acknowledging the balance of the proposed method between convolution and Mamba for local and global feature extraction. Reviewer 29W9 's positive remarks on the systematic evaluations and Reviewer UGvW’s remarks on the clarity of the paper were really encouraging.

We appreciate Reviewer 4y9X, Reviewer UGvW, Reviewer 29W9, Reviewer MY4P and Area Chair ycx7 for providing us with the opportunity to clarify the methodology and compare with more state-of-the-art methods.

We responded all comments from all the reviewers point-by-point and also updated the manuscript accordingly. We hope our answers addressed all questions.

**Supporting Material:**

/attachment/f9a21d3d01434acc9b80916735bc9be606dfb166.pdf

---

### Comment · Area_Chair_ycx7 · 2025-03-10
**rebuttal discussion**

Dear Reviewers,

Please look at the authors' reply to your initial review. I encourage you to engage in discussion and consider the author replies, and update your rating and assessment (if justified) until the 14th of March.

Based on the initial review, the rebuttal discussion, and your final grades, I will write meta-reviews and propose a decision to the PC.

Thank you again for your very useful reviews.

---

### Meta-Review · Area_Chair_ycx7 · 2025-03-21

**Recommendation:** Accept (Poster)
**Confidence:** 4

**Metareview:**

The paper proposes a hybrid architecture between convolutional networks and state-space models, with reduced complexity and memory footprint.

On the plus side, the reviewers noted the effectiveness and usefulness of reducing memory consumption and number of operations; the quality of the presentation; the evaluation on multiple datasets;

On the negative side, in general, thorough discussion and explanation of the difference with previous Mamba derivatives (e.g. Vision mamba, VMamba, U-Mamba, Segmamba) is missing, and claims of novelty are not explicitely made with respect to previous work, making it difficult to evaluate.

It is acknowledged that the authors made an important effort in evaluating on three different datasets, a very positive point. In terms of models, while several benchmarks are provided, the paper is missing a few key comparison that need to be applied systematically across datasets. 1) Comparison with closest related offerings, that is a combination of convnet and SSM - here SegMamba is presented (and SegResMamba compares favorably in terms of computation/performance tradeoff), but the paper is missing comparisons with U-Mamba which is rightly mentioned in the introduction, or SwinUMamba. 2) Comparison with generalist strong baselines like nnUnet v2 with residual encoder, for all datasets

It is acknowledged that the time for rebuttal is short and therefore efforts to include additional models in the benchmark of BTCV are appreciated. However, reply to reviewer UGvW indicates that differences with SegMamba have been discussed in the revised discussion section, and I could find no substantial changes there.

No code is provided.

The rebuttal addressed many of the points raised by the reviewers and helped improve the manuscript.

Overall, while weaknesses highlighted above remain, this is a valuable contribution to balance computational complexity and subsequent energy demands with performance for medical imaging data. As AI keeps gaining foothold in clinical environments, it will be particularly important to pay attention to energy consumption concerns, and this kind of research should be encouraged.